# From $r$ to $Q^*$: Your Language Model is Secretly a Q-Function

**Rafael Rafailov\***
Stanford University
rafailov@stanford.edu

**Joey Hejna\***
Stanford University
jhejna@stanford.edu

**Ryan Park**
Stanford University
rypark@stanford.edu

**Chelsea Finn**
Stanford University
cbfinn@stanford.edu

## Abstract

Reinforcement Learning From Human Feedback (RLHF) has been critical to the success of the latest generation of generative AI models. In response to the complex nature of the classical RLHF pipeline, direct alignment algorithms such as Direct Preference Optimization (DPO) have emerged as an alternative approach. Although DPO solves the same objective as the standard RLHF setup, there is a mismatch between the two approaches. Standard RLHF deploys reinforcement learning in a specific token-level MDP, while DPO is derived as a bandit problem in which the whole response of the model is treated as a single arm. In this work we rectify this difference. We theoretically show that we can derive DPO in the token-level MDP as a general inverse Q-learning algorithm, which satisfies the Bellman equation. Using our theoretical results, we provide three concrete empirical insights. First, we show that because of its token level interpretation, DPO is able to perform some type of credit assignment. Next, we prove that under the token level formulation, classical search-based algorithms, such as MCTS, which have recently been applied to the language generation space, are equivalent to likelihood-based search on a DPO policy. Empirically we show that a simple beam search yields meaningful improvement over the base DPO policy. Finally, we show how the choice of reference policy causes implicit rewards to decline during training. We conclude by discussing applications of our work, including information elicitation in multi-turn dialogue, reasoning, agentic applications and end-to-end training of multi-model systems.

## 1 Introduction

Reinforcement Learning from Human Feedback (RLHF) has become the defacto method for aligning large language models (LLMs) with human intent due to its success in a wide range of applications from summarization (Stiennon et al., 2022) to instruction following (Ouyang et al., 2022). By learning a reward function from human-labeled comparisons, RLHF is able to capture complex objectives that are in-describedable in practice. Following the success of (Ziegler et al., 2020), numerous works have considered new algorithms for training and sampling from large models in various domains using techniques from reinforcement learning (RL). In particular direct alignment methods, such as Direct Preference Optimization (DPO) (Rafailov et al., 2023) have gained traction in recent months because of their simplicity (Zhao et al., 2023a; Azar et al., 2023). Instead of learning a reward function and then using RL, direct alignment methods use the relationship between reward functions and policies in the contextual bandit setting to optimize both simultaneously. Similar ideas have since been applied to vision language (Zhao et al., 2023b) and image generation models (Lee et al., 2023).

---

\* Denotes equal contribution

While such direct alignment methods are purported to work the same as classical RLHF approaches that use policy gradient algorithms like PPO (Schulman et al., 2017), fundamental differences remain. For instance, classical RLHF methods optimize token-level value functions with a sparse reward at the terminal state. DPO on the other hand, operates only in a contextual bandits setting, treating the entire response as a single arm. This is despite the fact that tokens are generated one at a time, and dense rewards are commonly known to be beneficial in the RL community. While direct alignment algorithms are interesting, at present it is unclear if they can be applied to sequences in the same way as the underlying RL algorithms used in typical RLHF pipelines.

In this work we rectify this difference by deriving DPO within the token-level MDP setting present in large language models using the usual form of binary preference-feedback. We then show that DPO training implicitly learns a token-level reward function, for which the language models logits define the optimal Q function, or expected total future reward. We then demonstrate that DPO is able to flexibly model any possible dense reward function within the token MDP.

Empirically, we use our theoretical derivations to justify three practical insights which we believe to be of use to the community. First, we show that despite being derived as a contextual bandit, the implicit rewards of a DPO model have a per-token interpretation. Second, we demonstrate that likelihood search over a DPO model is analogous to searching over a reward function during decoding as done by contemporary works (Liu et al., 2023b; Feng et al., 2024). Finally, we identify the choice of initial policy and reference distribution as being important in determining the trajectory of implicit rewards during training.

## 2 Related Work

The problem of aligning policies with human intent using preference feedback has been a long studied problem in reinforcement learning (Akrour et al., 2011; Wilson et al., 2012). While the primary focus of RLHF was originally in control (Christiano et al., 2017), following the success of Ziegler et al. (2020) it has recently been broadly adopted by the language modeling (Ouyang et al., 2022; Nakano et al., 2021; Stiennon et al., 2022; Bai et al., 2022a) and even vision communities (Black et al., 2023a; Lee et al., 2023). Most works in RLHF optimize a learned reward function, used only at the end of generation, with a policy graident style-method. Such approaches have been known to be unstable (Engstrom et al., 2020) and hard to scale, while at the same time theoretically existing at an unusual intersection between contextual bandits and RL. In response, several direct alignment methods (Rafailov et al., 2023; Azar et al., 2023; Zhao et al., 2023a) have been developed which simplify the RLHF pipeline by learning a policy from preference data without an intermediate reward function. Such methods however, derived solely as contextual bandits, leave several theoretical and practical questions unanswered which we seek to address.

First, though direct alignment methods treat the LLM as a bandit, prior works have demonstrated that it is possible to use dense rewards Zelikman et al. (2022); Chan et al. (2024); Pan et al. (2023) or even approximate dynamic programming (Snell et al., 2022). Moreover, using the regret-model of preferences (Knox et al., 2023; 2024), Contrastive Preference Learning (Hejna et al., 2024) is able to use direct alignment for general MDPs, instead of the specific token MDP used in RLHF. Our work shows how DPO can be interpreted as optimizing a per-token reward function, which in practice is restricted to the family of optimal advantage functions.

Second, if DPO does not learn a reward function, can we still use its reward or value? Prior works have considered using best-of-K (Mudgal et al., 2023) or tree search (Liu et al., 2023b) for alignment with a value function Kim et al. (2022); Li et al. (2017) or discriminator (Yang & Klein, 2021). Using the implicit reward, we show that likelihood search results in a similar solution for direct alignment.

Our work builds on foundational knowledge in maximum entropy RL (Ziebart, 2010) and inverse RL (Ziebart et al., 2008; Ng et al., 1999; Cao et al., 2021). In particular, we leverage the mapping between $Q$-functions and reward functions under a fixed policy as first done

in inverse RL by Garg et al. (2022). Related to our work, Nachum et al. (2017) uses similar derivations for reinforcement learning in control and Watson et al. (2023) does so for inverse RL from demonstration. Hejna & Sadigh (2024) exploit this relationship for RLHF. While related, these works still require an additional loop of reinforcement learning optimization, which we dispose of in our formulation of feedback learning for LLMs. In the LLM domain, Yu et al. (2024) uses pre-trained models as priors for Q-learning, while Cundy & Ermon (2023) considers a similar formulation for imitation learning. In this work instead, we formulate preference-based learning as Q-learning.

## 3 Preliminaries

In this section we first define the per-token MDP for large language models, and then describe how it relates to classic RLHF approaches and direct alignment algorithms, specifically DPO. We operate in the typical RLHF setting where we have a dataset $\mathcal{D} = \{(\mathbf{x}^{(i)}, \mathbf{y}^{(i)})\}_{i=1}^N$ of language prompts $\mathbf{x}$ and target answers $\mathbf{y}$, which can each individually be broken down into a sequence of tokens, for example $\mathbf{x} = (x_0, \ldots, x_m)$, from a fixed discrete vocabulary $\mathcal{A}$. Throughout this section we will use the $\mathbf{x}, \mathbf{y}$ notation for the contextual bandit framing where the entire response $\mathbf{y}$ is the action, but will use state $\mathbf{s}$ and action $\mathbf{a}$ notation from RL literature for describing sequences at the token-level.

### 3.1 The Token-level MDP for Large Language Models

We define the token level MDP as a tuple $\mathcal{M} = (\mathcal{S}, \mathcal{A}, f, r, \rho_0)$, where the state space $\mathcal{S}$ consists of all tokens generated so far (i.e. $\mathbf{s}_t = \{x_0, \ldots, x_m, y_0, \ldots, y_t\}$) and the action space is the vocabulary of tokens $\mathcal{A}$. The dynamics $f$ are the deterministic transition model between tokens $f(\mathbf{s}, \mathbf{a}) = \mathbf{s}|\mathbf{a}$, where $|$ is concatenation. The initial state distribution $\rho_0$ is a distribution over prompts $\mathbf{x}$, where an initial state $\mathbf{s}_0$ is comprised of the tokens from $\mathbf{x}$. In RLHF, the reward function is learned from human feedback over preferences between responses which we will denote using trajectories $\tau$ at the token level. As is typically done (Ziegler et al., 2020; Stiennon et al., 2022), we assume that preference trajectories start at the same state (initial propmpt) and end in a terminal state (**EOS** token), from which future rewards are zero. In this token level MDP, the corresponding Bradley-Terry preference model Bradley & Terry (1952); Christiano et al. (2017) is

$$p^*(\tau^w \succeq \tau^l) = \frac{\exp\left(\sum_{i=1}^N r(\mathbf{s}_i^w, \mathbf{a}_i^w)\right)}{\exp\left(\sum_{i=1}^N r(\mathbf{s}_i^w, \mathbf{a}_i^w)\right) + \exp\left(\sum_{i=1}^M r(\mathbf{s}_i^l, \mathbf{a}_i^l)\right)}. \tag{1}$$

which gives the probability that the "win" trajectory $\tau^w$ of length $N$ is preferred to the "loss" trajectory $\tau^l$ of length $M$. Now that we have defined the token level MDP, we can show how it relates to both classic and direct alignment RLHF methods.

### 3.2 The Classical RLHF Methods

Most classical RLHF approaches (Ziegler et al., 2020; Bai et al., 2022b; Ouyang et al., 2022) first learn a reward function from human feedback on prompt and response pairs $(\mathbf{x}, \mathbf{y}^w, \mathbf{y}^l)$, then optimize it with a policy gradient-based method like PPO (Schulman et al., 2017) with an entropy-bonus using the following KL-constrained RL objective

$$\max_{\pi_\theta} \mathbb{E}_{a_t \sim \pi_\theta(\cdot|\mathbf{s}_t)} \left[ \sum_{t=0}^T (r(\mathbf{s}_t, \mathbf{a}_t) + \underbrace{\beta \log \pi_{\text{ref}}(\mathbf{a}_t|\mathbf{s}_t)}_{\text{KL penalty}}) + \beta \mathcal{H}(\pi_\theta)|\mathbf{s}_0 \sim \rho(\mathbf{s}_0) \right] \tag{2}$$

where $\pi_{\text{ref}}$ is a reference policy, often resulting from supervised finetuning, from which the learned policy should not significantly deviate. However, in classic RLHF methods the reward function is learned as a contextual bandit with the preference model

$$p^*(\mathbf{y}^w \succeq \mathbf{y}^l) = \frac{\exp r(\mathbf{x}, \mathbf{y}^w)}{\exp r(\mathbf{x}, \mathbf{y}^w) + \exp r(\mathbf{x}, \mathbf{y}^l)}$$

and is thus only applied at the final timestep for the last action where $\mathbf{a}$ is **EOS**. In practice the actual reward used in the token-level PPO is

$$r(\mathbf{s}_t, \mathbf{a}_t) = \begin{cases} \beta \log \pi_{\text{ref}}(\mathbf{a}_t|\mathbf{s}_t), & \text{if } \mathbf{s}_{t+1} \text{ is not terminal} \\ r(\mathbf{x}, \mathbf{y}) + \beta \log \pi_{\text{ref}}(\mathbf{a}_t|\mathbf{s}_t), & \text{if } \mathbf{s}_{t+1} = \mathbf{y} \text{ is terminal} \end{cases} \tag{3}$$

in a maximum entropy formulation. This leads to an interesting contradiction where the reward function $r$ is treated like a bandit, but the actual RL value function and optimization is done per-token in practice.

### 3.3 Direct Preference Optimization

Unlike classical RLHF, DPO, as derived in Rafailov et al. (2023), stays entirely within the contextual bandits setting entirely and also uses the bandit-based preference model in section 3.2. To circumvent the need for an RL algorithm, DPO uses the well-known closed form solution to the KL-contextual bandit version of the RL problem posed in eq. (2) (Ziebart et al., 2008; Levine, 2018):

$$\pi^*(\mathbf{y}|\mathbf{x}) = \frac{1}{Z(\mathbf{x})} \pi_{\text{ref}}(\mathbf{y}|\mathbf{x}) e^{r(\mathbf{x}, \mathbf{y})}$$

where $\pi^*$ is the optimal policy and $Z(\mathbf{x})$ is the partition function that normalizes it. DPO rearranges this equation to solve for reward as $r(\mathbf{x}, \mathbf{y}) = \beta \log \pi^*(\mathbf{y}|\mathbf{x}) - \beta \log \pi_{\text{ref}}(\mathbf{y}|\mathbf{x}) - Z(\mathbf{x})$. Substituting this relationship into the standard binary cross-entropy loss function used for reward modeling yields the DPO loss equation as the partition function $Z(\mathbf{x})$ cancels from the Bradley Terry model.

$$\mathcal{L}_{\text{DPO}}(\pi_\theta; \pi_{\text{ref}}) = -\mathbb{E}_{(\mathbf{x}, \mathbf{y}^w, \mathbf{y}^l) \sim \mathcal{D}} \left[ \log \sigma \left( \beta \log \frac{\pi_\theta(\mathbf{y}^w \mid \mathbf{x})}{\pi_{\text{ref}}(\mathbf{y}^w \mid \mathbf{x})} - \beta \log \frac{\pi_\theta(\mathbf{y}^l \mid \mathbf{x})}{\pi_{\text{ref}}(\mathbf{y}^l \mid \mathbf{x})} \right) \right] \tag{4}$$

For brevity we use $\sigma$ to denote the logistic function. In the next section, we show how an alternative derivation of DPO can also cast its optimization within the token-level MDP.

## 4 Theoretical Insights

In this section we explore how DPO can theoretically be cast into the token-level MDP, and explore the consequences of doing so. First, we provide a token level derivation of DPO under the assumptions in section 3.1. Next, we show that even in the token MDP, DPO is able to fit any reward function in the multi-step Bradley Terry preference model eq. (1). Ultimately, this shows that DPO can potentially be used for more sequential optimization tasks, like multi-turn interactions or even multi-modal generation.

### 4.1 DPO as a Q-function in the Token Level MDP

**RL in the Token-level MDP.** While the original derivation of DPO relies on the fact that $Q^*(\mathbf{x}, \mathbf{y}) = r(\mathbf{x}, \mathbf{y})$, this relationship does not hold in the token-level MDP. To resolve this, we need to develop new mathematical results that will allow us to relate the reward function in the Token-level Bradley Terry model eq. (1) to the corresponding optimal policy $\pi*$. In the general maximum entropy RL setting, the fixed point solution of eq. (2) is given by (Ziebart, 2010) as

$$\pi^*(\mathbf{a}_t|\mathbf{s}_t) = e^{(Q^*(\mathbf{s}_t, \mathbf{a}_t) - V^*(\mathbf{s}_t))/\beta} \tag{5}$$

where $\pi^*(\mathbf{a}|\mathbf{s})$ is the optimal policy and $Q^*(\mathbf{s}, \mathbf{a})$ is the optimal Q-function which models the total future reward from $(\mathbf{s}, \mathbf{a})$ under $\pi^*$. The optimal value function $V^*$ is a function of $Q^*$,

$$V^*(\mathbf{s}_t) = \beta \log \sum_{\mathbf{a} \in \mathcal{A}} e^{Q^*(\mathbf{s}_t, \mathbf{a})/\beta} \tag{6}$$

such that the policy $\pi^*$ integrates to one. Unfortunately unlike in the bandits setting this relationship gives us no specific information about the reward function $r$ at a single state

action pair since the optimal policy optimizes for total future returns as estimated by $Q$. To do so, we will need to consider the relationship between $Q^*$ and $r$.

**From $r$ to $Q^*$.** The relationship between future returns and the current timestep is captured by the belmman equaitons which are satisfed by any valid Q-function. We write this below for the optimal policy $\pi^*$ under the reward $r$ with a KL divergence penalty:

$$Q^*(\mathbf{s}_t, \mathbf{a}_t) = \begin{cases} r(\mathbf{s}_t, \mathbf{a}_t) + \beta \log \pi_{\text{ref}}(\mathbf{a}_t|\mathbf{s}_t) + V^*(\mathbf{s}_{t+1}), & \text{if } \mathbf{s}_{t+1} \text{ is not terminal} \\ r(\mathbf{s}_t, \mathbf{a}_t) + \beta \log \pi_{\text{ref}}(\mathbf{a}_t|\mathbf{s}_t), & \text{if } \mathbf{s}_{t+1} \text{ is terminal} \end{cases} \tag{7}$$

We can then rearrange the bellman equation for the optimal $Q$-function in terms of the reward. This style of relationship was first explored by Garg et al. (2022) in imitation learning and later in Hejna & Sadigh (2024) for preference-based RL. However, these works *require* the use of a discount factor $\gamma < 1$ which is typically not used in RLHF. In the appendix we prove the following Lemma which shows that this relationship is indeed one-to-one in the token MDP as well.

**Lemma 1.** *Under mild assumptions, there is a bijection between reward functions $r(\mathbf{s}_t, \mathbf{a}_t)$ and corresponding optimal Q-functions $Q^*(\mathbf{s}_t, \mathbf{a}_t)$ in the token MDP.*

This leads us to a rather interesting conclusion – that an LLM is *always* the optimal soft Q-functions for *some* reward function in the token MDP. Consider any LLM which outputs logits $l_\theta$ and temperature parameter $\beta$. As is common practice, we take the sampling policy $\pi$ to be the softmax over tokens modulated by temperature parameter $\beta$ – which is precisely eq. (5) where $Q^* = l_\theta$ because the value optimal function $V^*$ is precisely $\beta \log Z(\mathbf{s}_t)$, normalizing the distribution. The corresponding reward function may not be smooth or well-behaved. Notably, the logits have a free parameter due to the softmax. While this free-parameter results in the same optimal policy per later arguments, it means the sequence of values may not be smooth. The question then becomes how to finetune the LLM such that it is the optimal Q-function for a reward function $r$ that aligns with human preferences. To do so, we will complete our derivation of DPO in the token MDP.

**DPO learns our best estimate of $Q^*$.** Now that we have established a bijection between $r$ and $Q^*$, we can derive a token-level version of DPO to align the implicit reward, induced by the $Q$ function represented by the language model, with that of the best estimate of reward, according to Bradley-Terry model in eq. (1). To do so, we need to represent the sum of rewards first in terms of the $Q$-function $Q^*$, and then in terms of the policy $\pi^*$. We complete the first step by inverting the Bellman equation in eq. (7) and substituting it into the sum of rewards over a trajectory $\tau = \{\mathbf{s}_1, \mathbf{a}_1, \ldots, \mathbf{a}_{T-1}, \mathbf{s}_T\}$.

$$\sum_{t=0}^{T-1} r(\mathbf{s}_t, \mathbf{a}_t) = \sum_{t=0}^{T-1} \left( Q^*(\mathbf{s}_t, \mathbf{a}_t) - \beta \log \pi_{\text{ref}}(\mathbf{a}_t|\mathbf{s}_t) - V^*(\mathbf{s}_{t+1}) \right) =$$

$$= Q^*(\mathbf{s}_0, \mathbf{a}_0) - \beta \log \pi_{\text{ref}}(\mathbf{a}_0|\mathbf{s}_0) + \sum_{t=1}^{T-1} Q^*(\mathbf{s}_t, \mathbf{a}_t) - V^*(\mathbf{s}_t) - \beta \log \pi_{\text{ref}}(\mathbf{a}_t|\mathbf{s}_t)$$

The equality follows from $V^*(\mathbf{s}_T) = 0$ and re-arranging the sum to isolate $t = 0$. As $V^*$ is written entirely in terms of $Q^*$ and $\beta$ per eq. (6), we have expressed the sum of return over the sequence just in terms of $Q^*$. Next, we exchange $Q^*$ for $\pi^*$. We can log-linearize eq. (5) as $\beta \log \pi^*(\mathbf{a}_t|\mathbf{s}_t) = Q^*(\mathbf{s}_t, \mathbf{a}_t) - V^*(\mathbf{s}_t)$. This is equivalent to stating that the language model probabilities are just the softmax over $l_\theta = Q^*$ with temperature $\beta$. Continuing from the above, with this substitution we get

$$= Q^*(\mathbf{s}_0, \mathbf{a}_0) - \beta \log \pi_{\text{ref}}(\mathbf{a}_0|\mathbf{s}_0) + \sum_{t=1}^{T-1} \beta \log \frac{\pi^*(\mathbf{a}_t|\mathbf{s}_t)}{\pi_{\text{ref}}(\mathbf{a}_t|\mathbf{s}_t)} = V^*(\mathbf{s}_0) + \sum_{t=0}^{T-1} \beta \log \frac{\pi^*(\mathbf{a}_t|\mathbf{s}_t)}{\pi_{\text{ref}}(\mathbf{a}_t|\mathbf{s}_t)}$$

where the final step results from adding and subtracting $V^*(\mathbf{s}_0)$ and applying the substitution again. Now, this representation for the sum of rewards in terms of the optimal policy can be directly substituted into the preference model in eq. (1), where the $V^*(\mathbf{s}_0)$ term will cancel just as $Z(\mathbf{x})$ did in the original DPO derivation assuming $\tau^w$ and $\tau^l$ start

at the same state $\mathbf{s}_0$, giving us the policy-induced preference model

$$p_{\pi^*}(\tau^w \succeq \tau^l) = \sigma \left( \sum_{t=0}^{N-1} \beta \log \frac{\pi^*(\mathbf{a}_t^w|\mathbf{s}_t^w)}{\pi_{\text{ref}}(\mathbf{a}_t^w|\mathbf{s}_t^w)} - \sum_{t=0}^{M-1} \beta \log \frac{\pi^*(\mathbf{a}_t^l|\mathbf{s}_t^l)}{\pi_{\text{ref}}(\mathbf{a}_t^l|\mathbf{s}_t^l)} \right). \tag{8}$$

To derive the final DPO loss function, we can take the KL-divergence between the empirical preference model of our dataset $p_{\mathcal{D}}$ and the preference model implied by a learned policy $p_{\pi_\theta}$, $\mathbb{D}_{\text{KL}}(p_{\mathcal{D}}||p_{\pi_\theta})$. This results in

$$\mathcal{L}(\pi_\theta, \mathcal{D}) = -\mathbb{E}_{(\tau_w, \tau_l) \sim \mathcal{D}} \left[ \log \sigma \left( \left( \sum_{t=0}^{N-1} \beta \log \frac{\pi^*(\mathbf{a}_t^w|\mathbf{s}_t^w)}{\pi_{\text{ref}}(\mathbf{a}_t^w|\mathbf{s}_t^w)} \right) - \left( \sum_{t=0}^{M-1} \beta \log \frac{\pi^*(\mathbf{a}_t^l|\mathbf{s}_t^l)}{\pi_{\text{ref}}(\mathbf{a}_t^l|\mathbf{s}_t^l)} \right) \right) \right] \tag{9}$$

In the next section we demonstrate that DPO can learn any dense reward function in the token-level MDP.

## 4.2 Token-Level DPO Can Parameterize Any Dense Reward Function.

In the previous section we derived DPO using the bijection between reward functions and optimal $Q$-functions uniquely available in the token-level MDP. An alternative view of DPO casts it as restricting the learned reward function such that it belongs to the class optimal advantage functions $A^*(\mathbf{s}, \mathbf{a}) = Q^*(\mathbf{s}, \mathbf{a}) - V^*(\mathbf{s})$ from which an optimal policy is readily obtained per eq. (5). Here we show that this restriction does not limit the class of reward functions we can represent. We begin by expanding the definition of equivalency used in Rafailov et al. (2023) to the broader class of potential-based reward shaping functions:

**Definition 1.** *Two reward functions $r(\mathbf{s}_t, \mathbf{a}_t)$ and $r'(\mathbf{s}_t, \mathbf{a}_t)$ are equivalent if there exists a potential function $\Phi(\mathbf{s})$, such that $r'(\mathbf{s}_t, \mathbf{a}_t) = r(\mathbf{s}_t, \mathbf{a}_t) + \Phi(\mathbf{s}_{t+1}) - \Phi(\mathbf{s}_t)$.*

In Ng et al. (1999)'s seminal work, the authors proved that two equivalent reward functions defined per definition 1 have the same optimal policy. By log-linearizing the optimal policy fixed point in eq. (5) and substituting in the Bellman equation from eq. (7) (Nachum et al., 2017; Watson et al., 2023), we have

$$\beta \log \frac{\pi^*(\mathbf{a}_t|\mathbf{s}_t)}{\pi_{\text{ref}}(\mathbf{a}_t|\mathbf{s}_t)} = r(\mathbf{s}_t, \mathbf{a}_t) + V^*(\mathbf{s}_{t+1}) - V^*(\mathbf{s}_t). \tag{10}$$

This is precisely the optimal advantage function, where $V^*$ directly follows the form of a potential shaping function. Watson et al. (2023) first used this derivation to arrive at a "coherent" reward function and follow-ups arrived at the same conclusion by noting that using the advantage as reward preserves the optimal policy (Knox et al., 2024; Hejna et al., 2024). Unlike prior works, however, we demonstrate that this re-parameterization also leads to the same exact *preference* distribution as $r$.

**Theorem 1.** *Given a reference policy $\pi_{ref}$ and a parameter $\beta > 0$ all reward classes consistent with the Plackett-Luce (and Bradley-Terry) models in eq. (1) can be represented with the a re-parameterization of the form*

$$r(\mathbf{s}, \mathbf{a}) = \beta \log \pi(\mathbf{a}|\mathbf{s}) - \beta \log \pi_{ref}(\mathbf{a}|\mathbf{s}) \tag{11}$$

*within the token MDP where $V^*(\mathbf{s}_t) = 0$ for all terminal states.*

*Proof.* Above we derived the invariance of the optimal policy under the re-parameterization. The preference model can be shown to be invariant by substituting and following the same steps used to arrive at eq. (8) in the last section, or by following Definition 1 from Watson et al. (2023). □

Interestingly, in practice, the potential function $\Phi(\mathbf{s}_t)$ represents the free parameter in the logits of the language model. An equal shift along all logits yields the same policy, but different Q-functions and corresponding rewards. The above Theorem proves that all of these are in the same equivalence class and induce the same set of preferences.

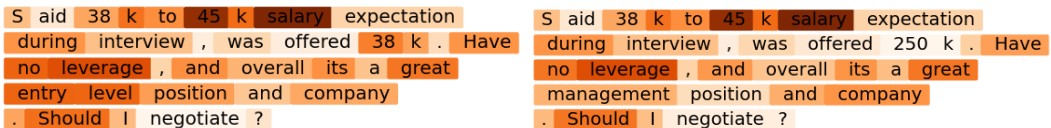

Figure 1: Credit assignment in DPO based on answer-level feedback. We provide two summaries to a Reddit post about a job interview. The left is the base response and on the right we have introduced errors in the salary range and the position level. Each token is colored corresponding to the DPO implicit reward as expressed in Eq. 11 (darker is higher), using the trained model. We see that the model correctly highlights the erroneous statements, without much change to the value of the other tokens, which indicates the ability to do credit assignment.

Moreover, this Theorem implies that we can use DPO to learn the optimal policy for any per-token reward function, provided preference queries start at the same state and end at a terminal state. In addition, DPO *always* fits an optimal advantage function for *some* reward which is responsible for credit assignment. Thus, the training data determines how close the learned advantage corresponds to that of the true reward. This is in contrast to methods that estimate the reward function and then additionally employ some policy improvement mechanism. Which algorithm performs better remains largely an open or empirical question.

The above derivations cast a language model as a Q function in the discrete token-level MDP. While this interpretation does not generally hold in continuous spaces, we can extend many of our results to other specially structured MDPs, like those present in diffusion. See Appendix B for more thorough treatment.

## 5 Practical Insights

In this section we discuss the empirical implications of our theoretical analysis. First, we qualitatively show that DPO can learn per-token credit assignment. Next, we use the derivations of the prior section to connect guided decoding and search-based algorithms, such as MCTS, to likelihood-based search on the DPO policy and empirically validate these results. Finally, (for the first time), we mathematically explain the phenomenon of decreasing likelihoods during DPO training, observed in the research and industry community.

For all empirical evaluations we use the Pythia 2.8B model Biderman et al. (2023) and the Reddit TL;DR summarization dataset Stiennon et al. (2022). We use the default hyper-parameters from the original public DPO implementation, unless otherwise stated.

### 5.1 Does DPO Learn Credit Assignment?

In the previous section we outlined how the trained DPO policy represents an optimal Q-function for some reward that optimizes the preference equation. In this section, we evaluate qualitatively if the DPO-trained model is able to learn credit assignment from trajectory feedback. We begin with a generic set of Reddit posts for the TL;DR test dataset, which we provide in Appendix C with additional examples. In our representative example the user discusses an employment negotiations situation. Two answers are shown in Figure 1. The base summary, which is correct is provided on the left. On the right we modify the summary by introducing a higher-level position and a corresponding higher salary. For each token in both answers we compute the DPO reward (equivalently the advantage function or "coherent" reward (Watson et al., 2023)), $r(\mathbf{s}, \mathbf{a}) = \beta \log \pi_\theta(\mathbf{s}|\mathbf{a}) - \beta \log \pi_{\text{ref}}(\mathbf{s}|\mathbf{a})$, where $\pi_\theta$ as outlined in Theorem 1 (here $\pi_\theta$ is our DPO-trained model and $\pi_{\text{ref}}$ is the SFT model). In Figure 1 each token is colored proportionally to this reward. We see that the model successfully identifies the tokens corresponding to the erroneous statements, while still maintaining comparable values for the rest, which is indicates that it can do credit assignment. Moreover, we see that within the context of the first error ("250K" salary) the model still allocates reasonable values to the rest of the tokens and specifically identifies the second error "management position". This is a promising sign of the ability to do "stitching"

Levine et al. (2020) i.e. a form of combinatorial generalization from offline data. If this is the case, our findings could be significant for the use of reinforcement learning and RLHF in LLMs, particularly for compositional tasks, such as code and reasoning. At the same time, in the recently introduced RewardBench Lambert et al. (2024), DPO models have demonstrated strong performance as classifiers on reasoning tasks. We believe these are encouraging results, which warrant further large-scale study beyond our qualitative observations.

## 5.2 Connecting Guided Decoding and Search to Likelihood-Based DPO Optimization

Recently Large Language Models have been combined with search algorithms during the inference stage Mudgal et al. (2024); Feng et al. (2024); Huang et al. (2024); Liu et al. (2023a), which have found to improve the quality of responses over standard next token decoding. Following the standard literature, these methods rely on a (usually sparse) reward signal or model $r_\theta(\mathbf{s_t}, \mathbf{a}_t)$ which they use to train a separate value function $V_\theta(\mathbf{s}_t)$. During inference time they deploy a graph-search algorithm in the token MDP as outlined in Section 3.1 to maximize the sum of rewards. Let us consider the search problem outlined in Eq. 2 with a partial expansion of length $K$:

$$\max_{\mathbf{a}_0,\dots,\mathbf{a}_K} r(\mathbf{s}_0, \mathbf{a}_0) + \beta \log \pi_{\text{ref}}(\mathbf{s}_0, \mathbf{a}_0) + \dots + r(\mathbf{s}_t, \mathbf{a}_t) + \beta \log \pi_{\text{ref}}(\mathbf{s}_K, \mathbf{a}_K) + V^*(\mathbf{s}_{K+1}) \quad (12)$$

where $V^*$ is the optimal corresponding value function. Now, if we directly substitute the reward representation from Eq. 10 into the above and considering a telescoping sum, with some standard algebra, we obtain that the above objective is equivalent to

$$\max_{\mathbf{a}_0,\dots,\mathbf{a}_K} -V^*(\mathbf{s}_0) + \beta \log \pi^*(\mathbf{a}_0|\mathbf{s}_0) + \dots + \beta \log \pi^*(\mathbf{a}_K|\mathbf{s}_K) \quad (13)$$

where $\pi^*$ is the corresponding optimal policy. Now, since the starting state is fixed (it's given by the prompt) we have that a search algorithm based on the conservative reward function of the RLHF objective and the corresponding optimal value policy is equivalent to likelihood search on the corresponding optimal policy. We empirically verify this property in Fig. 2, which shows the win rate of DPO models trained with three different $\beta$ values against the preferred summary in the test dataset. We see that a 5-beam search improves win-rates by 10-15% over the base policy (1-beam), which is comparable to the value-function guided search improvements reported in Mudgal et al. (2024). Interestingly, we see performance degrade with higher number of beams and that answers with exploding length are produced, which is a sign of reward over-optimization Gao et al. (2023); Park et al. (2024); Rafailov et al. (2024) and would explain the degradation in performance. These observations are consistent with our formulation of beam search as a search over a learned reward function.

These findings are consistent with the result of the recently proposed V-STaR algorithm Hosseini et al. (2024), which combines the approach of STaR Zelikman et al. (2022) with a DPO trained verifier. At inference time, the STaR model produces several candidate reasoning chains (plans) which are ranked by the DPO verifier likelihood. This can be seen as a form of likelihood based search as in Eq. 12, however instead of directly searching on the DPO model, it uses the STaR model as a proposal distribution. We hypothesize this is beneficial in preventing reward hacking, which is potentially an issue with deeper search as shown in Fig. 2.

## 5.3 Connections Between Proxy Tuning and Reinforcement Learning

Several recent works Mitchell et al. (2023); Liu et al. (2024a;b) have proposed an approach of inference-time model alignment through a proxy guidance model. These approaches start with a (unaligned) base model $\pi_{\text{base}}$ and a proxy model $\pi_{\text{proxy}}$ and a target distribution reference model $\pi_{\text{ref}}$. The inference time re-alignment of the base model is carried by re-weighting the conditional probabilities of each token:

$$\pi(\mathbf{a}|\mathbf{s}_t) \propto \pi_{\text{base}}(\mathbf{a}|\mathbf{s}_t) \left( \frac{\pi_{\text{proxy}}(\mathbf{a}|\mathbf{s}_t)}{\pi_{\text{ref}}(\mathbf{a}|\mathbf{s}_t)} \right)^\beta \quad (14)$$

Under our considerations from the prior chapter, then this becomes equivalent to

$$\pi(\mathbf{a}|\mathbf{s}_t) \propto \pi_{\text{base}}(\mathbf{a}|\mathbf{s}_t) \exp(\beta(Q^*(\mathbf{s}_t, \mathbf{a}) - V^*(\mathbf{s}_t))) \quad (15)$$

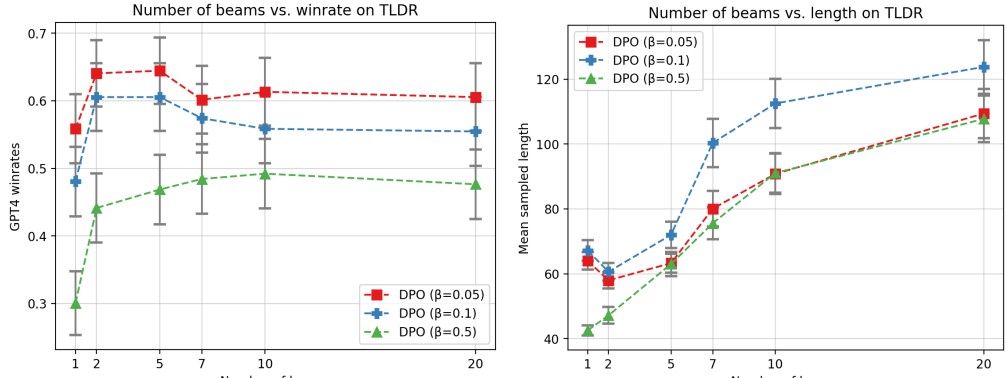

Figure 2: Model performance using beam search. **Left**: Win rate of the model generated summaries over the preferred summary on 256 held-out test prompts from the Reddit TL;DR dataset, as evaluated by GPT 4. **Right**: The average answer length based on number of beams. We see exploding verbosity with more than 5 beams, which also leads to lower model win rates, despite GPT4's well-know preference length bias.

where $\beta(Q^*(\mathbf{s}_t, \mathbf{a}) - V^*(\mathbf{s}_t))$ is the optimal implicit advantage from the proxy tuning model. That is our theoretical results allows us to tie the realignment approaches of Mitchell et al. (2023); Liu et al. (2024a;b) to recent works which explicitly train critic models Mudgal et al. (2024) for token-level decoding.

## 5.4 Likelihoods should decrease when using DPO.

A surface level interpretation of DPO would lead one to believe it increases the likelihood of chosen responses, while decreasing the likelihood of rejected responses. This however, does not account for a well observed phenomena in which the likelihood of the chosen responses actually *decrease* over time (Pal et al., 2024). This is illustrated on the left half of fig. 3, which we show that when performing SFT before DPO, the implicit rewards of both the chosen and rejected response decline, though the margin between them increases. However, given a MaxEnt RL framing, this phenomena may be expected.

Consider the expected log ratio (or implicit / "coherent" reward) of a policy under the reference model, which is often measured during training. Algebraic manipulation yields the following relationship:

$$\mathbb{E}_{\mathbf{a}\sim\pi_{\text{ref}}(\cdot|\mathbf{s})}\left[\beta\log\frac{\pi(\mathbf{a}|\mathbf{s})}{\pi_{\text{ref}}(\mathbf{a}|\mathbf{s})}\right] = -\beta\mathbb{D}_{\text{KL}}\left(\pi_{\text{ref}}(\cdot|\mathbf{s})||\pi(\cdot|\mathbf{s})\right) \tag{16}$$

At the beginning of training when $\pi = \pi_{\text{ref}}$, the implicit rewards are trivially zero. However at the end of training, assuming $\pi_{\text{ref}} \neq \pi^*$, the KL-divergence is necessarily positive, indicating that the implicit rewards must decrease in expectation to converge. This means that the average implicit rewards *should* go down when starting from the SFT model. In fact, on the left side of fig. 3 we show that when one *does not* SFT before DPO, there is little discernible trend in the average implicit reward and the implicit rewards of the chosen responses remain above zero. In fact, this trend also holds for CPL Hejna et al. (2024) for the general MDP, where the implicit rewards actually increase if SFT is not used.

One might realize that the previous analysis does not necessitate that the implicit rewards of the chosen must decrease, just that the implicit rewards must decrease on average. However, in practice it is common place (and recommended by Rafailov et al. (2023)) to SFT on only the chosen responses to form $\pi_{\text{ref}}$. For this section only we will call this choice of reference $\pi_{\text{ref}}^w$. Substituting $\pi_{\text{ref}}^w$ into eq. (16), we can see that when SFTing on the positive answers the implicit rewards of the chosen responses *must* go down because at convergence as $\mathbb{E}_{\pi_{\text{ref}}^w}[\beta\log\pi^* - \beta\log\pi_{\text{ref}}^w] = -\beta\mathbb{D}_{\text{KL}}(\pi_{\text{ref}}^w||\pi^*)$.

**Based on this derivation and choice of $\pi_{\text{ref}}^w$, the likelihood of the chosen response should decrease in the process of DPO training.**

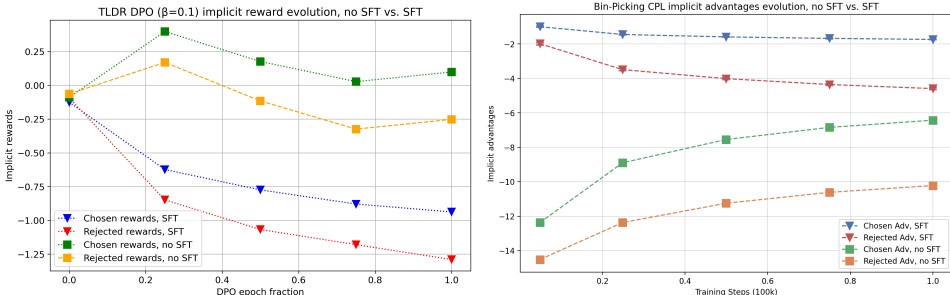

Figure 3: The evolution of implicit rewards for DPO on TLDR (left) and CPL on the bin-picking dataset (right) during training. We see that when we start with SFT, reward values decrease, whereas starting without SFT causes implicit rewards to be positive for DPO and increase for CPL.

While choosing $\pi_{\text{ref}} = \pi_{\text{ref}}^{w}$ is done in practice (Rafailov et al., 2023), it does mean that DPO will decrease the likelihood of all data in favor of extrapolated responses, which could cause over-fitting. Moreover, now that we have provided a derivation of DPO in the token-level MDP, one might expect it to exhibit characteristics like an RL algorithm – namely that the implied $Q$-function monotonically increases over time. However, this is not necesarily the case. Note that per analysis in Section section 3.1, DPO can be viewed as adjusting the reward (or advantage) from which the optimal policy is deterministically mapped within the token-level MDP. DPO does not train a policy to maximize reward, and thus we do not argue about whether its implied value functions should increase or decrease over time.

## 6  Discussion

In this work we formulated the DPO optimization algorithm as learning an optimal Q-function, which is represented by an LLM. This formulation and our results provide theoretical justification for empirically observed DPO training phenomena, which are not explained by the original bandit formulation. We further link and unify a family of proposed new LLM search algorithms by likelihood search under DPO and show comparable empirical gains by a simple 1-line code change to using beam search. Most importantly, we show qualitative early signs that DPO is able to learn credit assignment directly from feedback data. While larger-scale empirical exploration is necessary, we believe this an encouraging early sign. Our results indicate a number of promising future directions to explore:

**Learning intermediate reasoning from outcome feedback:** Recent works have shown promising results on that front Pang et al. (2024); Hwang et al. (2024).

**Multi-turn conversations:** Teaching language models to be an interactive conversationalists has been difficult, as RLHF is optimized as a single-turn bandit formulation. Moreover, classical methods, such as PPO, are not applicable in this setting. Recent work by Andukuri et al. (2024) has shown success in this domain using STaR and extending DPO to multi-turn conversational trees is a promising direction.

**Agentic LLMs:** LLM agents, such as WebGPT (Nakano et al., 2022) are equipped to take autonomous actions, such as browsing the Web and collecting information before providing an answer. The user then provides feedback based on the final output. Our derivations indicate that DPO training (on the full model trajectories) could learn optimal exploration behaviour. Recent works Song et al. (2024); Xi et al. (2024) shows promise in that direction.

**End-to-end training of generative AI systems:** Modern image generation systems, such as Dalle 3 Betker et al. (2023) use an LLM to produce high quality conditioning before calling a diffusion generation model. Also, recent long-form video generation models Hu et al. (2023); Gupta et al. (2023) combine transformer-based auto-regressive generations with a diffusion-based decoder. Such systems could potentially be optimized end-to-end with a hybrid version of DPO. We expand on these points in the Appendix.

We believe these are promising directions for future work.

**Acknowledgements**

Chelsea Finn is a CIFAR Fellow in the Learning in Machines and Brains program. JH is supported by an NDSEG Fellowship. This work was also supported by ONR grant N00014-22-1-2621 and the Volkswagen Group.

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

# A  Proof of Lemma 1

**Lemma 1.** *For a fixed policy $\pi$, there is a bijection between reward functions $r$ and corresponding optimal Q-functions ($Q^*$) in the deterministic tree-structured LLM MDP.*

*Proof.* Let $Q_r^*$ denote the optimal $Q$-function for reward $r$. We prove the statement directly, starting with the injective case.

Assume there exists a reward function $r' \neq r$ such that $Q_{r'}^* = Q_r^*$. Then, there must exist a state action pair such that $r'(\mathbf{s}_t, \mathbf{a}_t) \neq r(\mathbf{s}_t, \mathbf{a}_t)$. In fact, proceeding backwards from a leaf node (terminal state), there must be a *first* state action pair $(\mathbf{s}_t, \mathbf{a}_t)$ where $r'(\mathbf{s}_t, \mathbf{a}_t) \neq r(\mathbf{s}_t, \mathbf{a}_t)$. The $Q$ functions at this location are

$$Q_{r'}^*(\mathbf{s}_t, \mathbf{a}_t) = r'(\mathbf{s}_t, \mathbf{a}_t) + V_{r'}^*(\mathbf{s}_{t+1}), \quad Q_r^*(\mathbf{s}_t, \mathbf{a}_t) = r(\mathbf{s}_t, \mathbf{a}_t) + V_r^*(\mathbf{s}_{t+1})$$

By the fact that this was the *first* location where the reward functions differed starting from a leaf node, we must have that $V_{r'}^*(\mathbf{s}_{t+1}) = V_r^*(\mathbf{s}_{t+1})$. This is because we can recursively solve for the optimal policy, value, and Q-function using eq. (5) eq. (7), and eq. (6) from Ziebart et al. (2008). The rewards in all possible future states from $s, a$ are equal by virtue of this being the location of the first difference and thus the dynamic programming solution up to this point is the same. Thus, we can see that $Q_{r'}^*(\mathbf{s}_t, \mathbf{a}_t) \neq Q_r^*(\mathbf{s}_t, \mathbf{a}_t)$, completing this direction. Note that this proof does not hold in general MDPs, only the token MDP where it is impossible to return to the same state after taking any number of actions.

The surjective direction is easier. For all $Q^*$, we can compute a reward function $r(\mathbf{s}_t, \mathbf{a}_t) = Q^*(\mathbf{s}_t, \mathbf{a}_t) - V^*(\mathbf{s}_{t+1})$ under deterministic dynamics. Thus, we can see that the mapping is surjective.

## B  Treatment of Diffusion Models

Conditional diffusion image generation models, such as Stable Diffusion 3 Esser et al. (2024) have also used a form of the DPO algorithm as outlined in Wallace et al. (2023). Our analysis can no longer be directly applied in that setting, since the generations are continuous. However, we could translate many of our results to that setting, if we consider a certain diffusion MDP. We outline our results bellow.

### B.1  Diffusion MDP

We borrow the denoising MDP formulation from Black et al. (2023b); Fan et al. (2023). We again have the tuple $(\mathcal{S}, \mathcal{A}, f, r, \rho_0)$, with the same formulation as in Section 3.1. At the same time consider a diffusion process with time index $t$ and $T$ total steps, conditioned on context $\mathbf{c}$ and image denoted by $\mathbf{x}_t$. Then we can map the diffusion generation process to an MDP in the following way

$$\mathbf{s}_t = \begin{cases} (\mathbf{c}, T) & \text{if } t = 0 \\ (\mathbf{c}, \mathbf{x}_{T-t}, T - t) & \text{otherwise} \end{cases}$$

That is the initial state consists of the prompt $\mathbf{c}$ and afterwards each state consists of the current denoised image $\mathbf{x}_t$ and time step $T - t$. Notice that the time-steps in the MDP are inverse to the direction of the diffusion process (i.e. we start at noise and end at the final image). The action is just the next image iteration, from where the dynamics is also straightforward:

$$\mathbf{a}_t \triangleq \mathbf{x}_{T-t+1}$$
$$f(\mathbf{s}_t = (\mathbf{c}, \mathbf{x}_{T-t}, T - t), \mathbf{a}_t) = (\mathbf{c}, \mathbf{a_t}, T - t - 1)$$

Notice that in this case the policy is **stochastic**, but the dynamics of the MDP is still **deterministic**. Finally, the initial distribution, is just the distribution of prompts:

$$\rho(\mathbf{s}_0) \triangleq (p(\mathbf{c}), 0)$$

### B.2  Theoretical Results for the Diffusion MDP

Given the above formulation, we can also prove that Lemma 1 also holds in the diffusion MDP.

**Lemma 2.** *Under mild assumptions, there is a bijection between reward functions $r(\mathbf{s}_t, \mathbf{a}_t)$ and corresponding optimal Q-functions $Q^*(\mathbf{s}_t, \mathbf{a}_t)$ in the diffusion MDP.*

*Proof.* Since the MDP still has deterministic dynamics, we have that Eq. 5-7 still hold. Now, given a reference policy $\pi_{\text{ref}}$, parameter $\beta$ and a critic $Q$, we can trivially recover the unique reward function by inverting Eq. 7. We will prove that given a reward function $r(\mathbf{s}_t, \mathbf{a}_t)$, we can recover a unique critic $Q$. We work inductively in the diffusion MDP starting with $t = T$, where we have $V^*(\mathbf{s}_T) = 0$ for all terminal states. We then have that

$$Q^*(\mathbf{s}_{t-1}, \mathbf{a}_{t-1}) = Q^*(\mathbf{s}_t = (\mathbf{c}, \mathbf{x}_{T-t+1}, T - t + 1), \mathbf{a} = \mathbf{x}_{T-t}) =$$
$$r(\mathbf{s}_t = (\mathbf{c}, \mathbf{x}_{T-t+1}, T - t + 1), \mathbf{a} = \mathbf{x}_{T-t}) + \beta \log p_{ref}(\mathbf{x}_{T-t} | \mathbf{c}, \mathbf{x}_{T-t+1}, T - t + 1) +$$
$$\beta \log \int_{\mathcal{A}} e^{Q^*(\mathbf{s}_t = (\mathbf{c}, \mathbf{x}_{T-t}, T-t), \mathbf{x}_{T-t-1})/\beta} d\mathbf{x}_{T-t-1}$$

where $\pi_{ref}$ is the reference backwards diffusion process. In this case even though the state space is deterministic, our approach to the proof of Lemma 1 still holds by using backwards induction on the diffusion step $t$. Notice, that from $V(\mathbf{s}_T = (\mathbf{c}, \mathbf{x}_0, 0)) = 0$ we can uniquely determine the critic values for all states at time step $T - 1$. Proceeding inductively backwards through time in the MDP/denoising process (forward in the diffusion process), we obtain the desired result. $\square$

Given the proof of this Lemma, we can then directly apply the results of Section 4.2, including Theorem 1. Our results, also give us insights into the formulation of Wallace et al. (2023). In particular, by changing the sampling scheme of the intermediate diffusion the authors obtain two alternative formulations (Appendix S2 in Wallace et al. (2023)). Both of these schemes are suggested as empirical approximations in the formulation of the Diffusion-DPO algorithm, however in the view of Q-learning both of these are valid approaches to generating off-policy data. In fact, our interpretation of DPO allows for general off-policy data sampling and aggregation methods, which could yield a whole family of DPO algorithms in this domain. We leave the exploration of this direction for further work.

## C   Reddit TL;DR Posts

SUBREDDIT: r/running

TITLE: Tips on getting back into running after 4 years of not doing so & shin splints

POST: Hey everyone, I was hoping to gather some tips from people who left running and had to start over. A semi-lengthy background on myself to help you understand where I am coming from. In high school I was a very good cross country runner, running from 35-50 miles a week and never slower than 8-9 minute miles. At the end of senior year, I planned on taking a break from running and then try to race half or full marathons in the spring. I ended up not running at all after xc. 4 years later, I was noticing how much I miss the sport (especially after seeing the success of xc friends) so I decided to join a running group to get back into it. But the only group at my university that I could find was a triathlon club. I joined them, but only did the running workouts. After about 4 weeks, I developed shin splints. This is because I haven't ran in 4 years but thought 6 miles was ok after 4 weeks. Also, being 25 pounds heavier didnt help. After taking 3 months off and training on the bike and in the pool, I finally was back to running in february. but my shinsplints was still around. I finished my first sprint triathlon last week, and have been trying to get miles back under my feet again. I havent felt shin splints severely since the beginning of March, but I can feel it looming around. After a half year of it, I am getting really really frustrated. I cant run more than 4 miles still and my fastest mile is 8 minutes. I know I will probably never run like I did when I was 17, but its difficult because of remembering what I used to be capable of running.

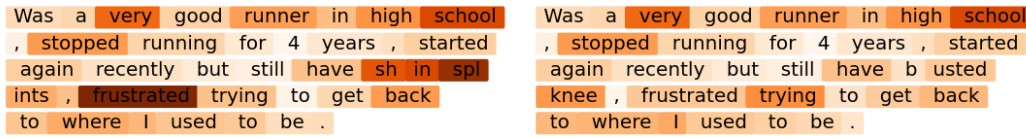

DPO successfully identifies the wrong tokens ("busted knee") in the right summary and correctly classifies this pair.

SUBREDDIT: r/AskReddit

TITLE: What is a sub-$800 camera that can shoot high quality video ideal for music video-like appearances?

POST: [This is a video of what we're trying to achieve.](

My school currently has a Sony HVR-HD1000u, and compared to that, our videos are nowhere near as good. I understand that things like lighting and color correction play a pretty big role, but even then I feel like our videos are never that clean. I usually can't get 720p clips out of our camera and the slow motion that they have is something we can't even come close to.

One possible *problem* is that for some reason we can't use firewire to connect the camera to the computer so we have to play the tape on this thing that basically plays it and then we capture the tape playing. I feel like this is probably a huge problem because it's like trying to show a friend a movie by screen-capping from Skype.

SO, should we scrap the HVR-HD1000u and get a Canon T2i (a cheaper DSLR which from the samples I've seen on YouTube and clips from that video, seems pretty high quality), or continue trying to use the Sony?

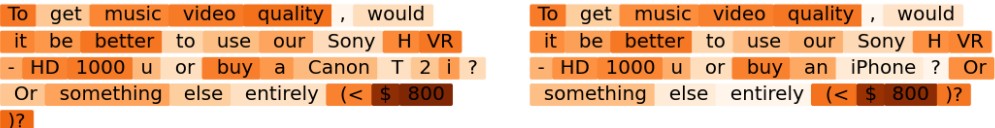

DPO successfully identifies the wrong tokens ("iPhone") in the right summary and correctly classifies this pair.

SUBREDDIT: r/personalfinance

TITLE: When asked about salary expectations during my interview I said 38k to 45k. Was just offered the position with 38k. Should I try and negotiate?

POST: So I interviewed for a position last week, and before the interview I saw online that the industry average for this position was $41,000. During the interview, they asked me my salary expectations, I said between $38,000 and $45,000 hoping it'd land somewhere in the middle. I received my offer today, and it was for $38,000. I can't help but wonder if I had just said $41,000 they probably would've offered it...

Anyways, so what I know is they are hiring 3 other people for this same position... I either got lucky and guessed exactly what salary they were planning on paying all of us to begin with, or we're all getting paid differently. As for the job, it is the ideal entry level position for me right now, and is a great company with benefits etc so I actually wouldn't mind working there for the 38k salary.

But it would be nice to get an even 40 at least, so my question is, is it common practice to negotiate salary after receiving an offer already? I also must say that I don't have any leverage as this is entry level and I would have probably still accepted had the offer been even as low as 30k. As such, I'm very afraid the offer may be retracted if I do try and negotiate, if that sort of thing happens?

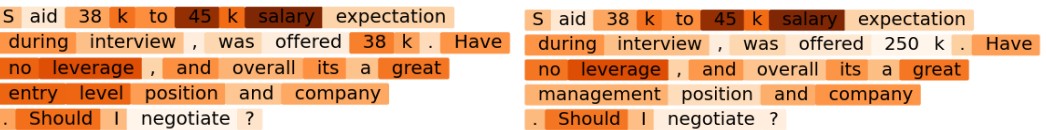

DPO successfully identifies the wrong tokens ("250k" and "management position") in the right summary and correctly classifies this pair.

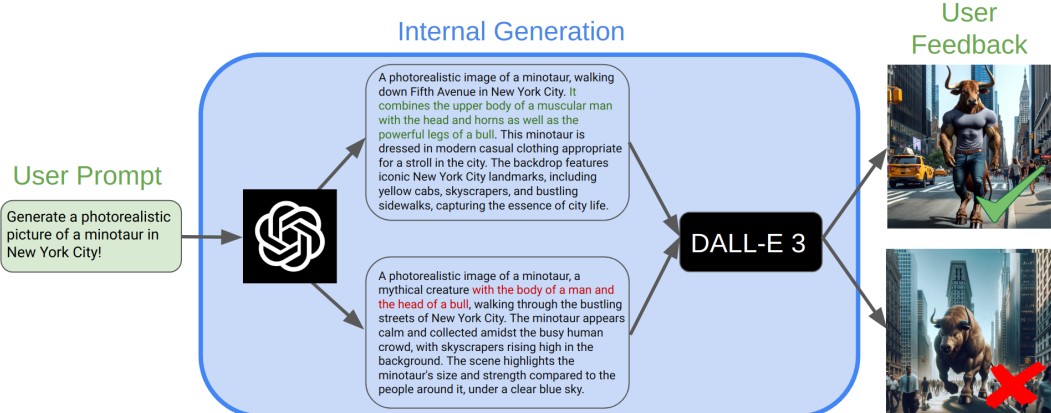

Figure 4: Example of an end-to-end generative AI workflow. The user request an image of a Minotaur in the streets of New York. However, we see that the rejected image does not actually represent a Minotaur, but a bull. The prompt refiner generates a valid description and specifically includes the phrasing "the body of a man and the head of a bull", but the image generation model fails to follow that prompt. At the same time in the case of the chosen image (which does reflect the prompt) the prompt refiner generates a more descriptive text, which the image generator is able to follow more closely. While it is possible to train each component separately, joint training can in theory, optimize the refiner to generate prompts that the image generator can more closely realize, while also training the generator to directly produce more aligned images.

## D   End-to-End Training of Generative AI Systems

**End-to-end system:** Here, we present an outline for end-to-end training of generative AI systems from human feedback using DPO in multi-step MDP. We assume two models - a prompt refiner $\pi_\theta(\mathbf{z}|\mathbf{x})$, which is a language model, generating discrete tokens in an autoregressive way, which, given a prompt $\mathbf{x}$ produces a refined prompt $\mathbf{z}$. This prompt is then fed into an image-generation diffusion model $\pi_\phi(\mathbf{y}|\mathbf{z})$, (which we parameterize as a denoising model $\epsilon_\phi$), which generates the final image. A real case example of that system is shown in Figure 4.

**Feedback generation:** During the feedback gathering stage, a user provides a query $\mathbf{x}$ and two refined prompts are sampled from $\mathbf{z}^1, \mathbf{z}^2 \sim \pi_\theta(\mathbf{x}|\mathbf{z})$, which the user does not directly evaluate. Then, the image generation model generates two images $\mathbf{y}^i \sim \pi_\phi(\mathbf{y}|\mathbf{z}^i)$ for $i = 1, 2$. The user then provides a preference over the $\mathbf{y}^i$ to yield the preference pair $\{\mathbf{y}^w, \mathbf{z}^w \succ \mathbf{y}^l, \mathbf{z}^l|\mathbf{x}\}$

**Optimization:** Optimization is carried in a hybrid MDP, where the initial state is the prompt $\mathbf{x}$ and has the same form as the token MDP, as outlined in 3.1. When the **EOS** token is encountered in that MDP, at which point the transition dynamics switches to the diffusion denoising MDP introduced in Black et al. (2023a). Notice that this is still a valid MDP and all the conclusions of our derivations in the main section of our paper hold. We could then optimize this system end-to-end using a hybrid DPO objective, combining our results, with those presented in Wallace et al. (2023) we have:

$$r_{\theta,\phi}(\mathbf{x}^w, \mathbf{y}^w) = \beta \underbrace{\sum_{i=0}^{|\mathbf{z}^w|} \log \frac{\pi_\theta(\mathbf{z}_i^w|\mathbf{x}, \mathbf{z}_{<i}^w)}{\pi_{\text{ref}}(\mathbf{z}_i^w|\mathbf{x}, \mathbf{z}_{<i}^w)}}_{\text{prompt refiner MDP}} +$$

$$\gamma T \omega(\lambda_t) \mathbb{E}_{t \sim \mathcal{U}(0,T)} \left[ \underbrace{(\|\epsilon^w - \epsilon_{\text{ref}}(\mathbf{y}_t^w, \mathbf{z}^w, t)\|_2^2 - \|\epsilon^w - \epsilon_\phi(\mathbf{y}_t^w, \mathbf{z}^w, t)\|_2^2)}_{\text{diffusion MDP objective}} \right] \quad (17)$$

where $\beta$ and $\gamma$ are two separate discounting factors for each modality. Here the diffusion objective follows directly from the derivation of Eq. 9 and the sampling scheme proposed in Wallace et al. (2023) (Eq. 12-14 in that paper). Notice here that the image generation model is conditioned on the corresponding refined prompt $\mathbf{z}^w$. We can define $r(\mathbf{x}^l, \mathbf{y}^l)$ similarly and optimize the DPO objective:

$$\mathcal{L}_{\text{DPO}_{\theta,\phi}} = -\mathbb{E}_{(\mathbf{x},\mathbf{y}^w,\mathbf{y}^l)\sim\mathcal{D}} \left[ \log \sigma \left( r_{\theta,\phi}(\mathbf{x}^w, \mathbf{y}^w) - r_{\theta,\phi}(\mathbf{x}^l, \mathbf{y}^l) \right) \right] \tag{18}$$

We demonstrate a particular real use of such a system in Figure 4. The user request an image of a Minotaur in the streets of New York. However, we see that the rejected image does not actually represent a Minotaur, but a bull. The prompt refiner generates a valid description and specifically includes the phrasing "the body of a man and the head of a bull", but the image generation model fails to follow that prompt. At the same time in the case of the chosen image (which does reflect the prompt) the prompt refiner generates a more descriptive text, which the image generator is able to follow more closely. While it is possible to train each component separately, joint training can in theory, optimize the refiner to generate prompts that the image generator can more closely realize, while also training the generator to directly produce more aligned images.

### D.1 Hybrid Video Generative Models

A recent line of work on long-form video generation Hu et al. (2023); Gupta et al. (2023) by combining auto-regressive transformer generation with a diffusion model decoding or uspcaling of the actual video frames to obtain temporally consistent and high-fidelity generations. We could deploy similar RLHF pipelines to the video-generation problem as well. It is also straightforward to extend the DPO joint optimization framework, presented in the previous section in Eq. 18 to this stetting as well. Instead of textual prompt refiner tokens, the variables $\mathbf{z}$ would represent the latent token generations of the autoregressive component $\pi_\theta$, which would be decoded into actual video frames $\mathbf{y}$ via the diffusion decoder $\pi_\phi$. We believe this is an exciting direction to pursue for the emerging video generation technologies.

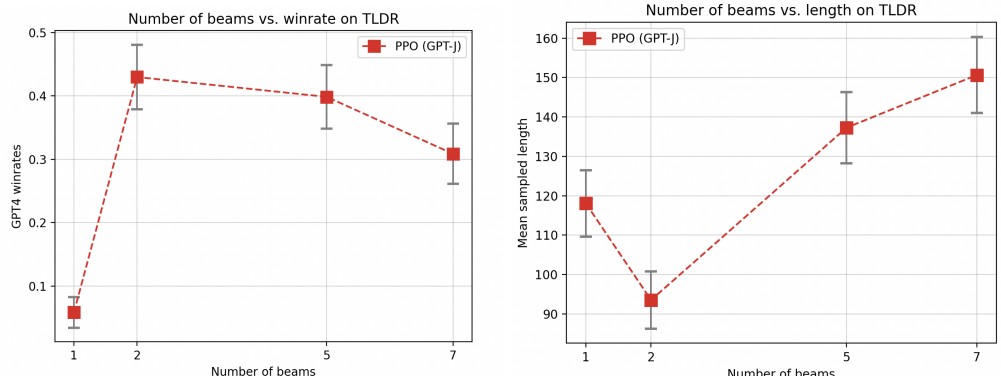

Figure 5: PPO model performance using beam search. **Left**: Win-rate of model summaries over preferred summary. **Right**: Average answer length based on number of beams.

## E    Beam Search Trends for PPO

We consider whether the beam search trends in Figure 2 hold for PPO, since PPO is known to exhibit reward over-optimization as well (Gao et al. (2023)). We use a PPO-tuned GPT-J-6B model released by CarperAI[1], fine-tuned on the same TL;DR dataset. We use the same sampling parameters as with the DPO experiments ($\tau = 1.0$, $k = 50$) and generate 256 samples from test set prompts, with GPT-4 as the evaluator. We only report results for 1, 2, 5, and 7 beams; higher number of beams were tried but exhausted memory on an NVIDIA A40 GPU.

In Figure 5, we observe a similar over-optimization phenomenon as with DPO, with 2 beams both increasing winrate and decreasing sample length (even under the length-biased evaluator GPT-4). However, more than 2 beams leads to a decline in downstream performance and large uptick in sample length, similar behavior to DPO in the over-optimization regime. We find that the benefits of increasing the beam size are more limited when using this model.

---

[1] https://huggingface.co/CarperAI/openai_summarize_tldr_ppo

## F    TL;DR Sample Generations

### F.1    Samples Across Different Beam Searches

All samples generated from the $\beta = 0.1$ checkpoint.

SUBREDDIT: r/AskReddit

TITLE: Question regarding learning/teaching.

POST: I apologize in advance if this is not the right sub reddit for this kind of question.

for the past 2 years, i've been slowly but surely developing my illustration skills. I am 18, recently out of high school, and will be going to college for a bachelors in illustration in likely 1-2 years.

Despite teaching my self to be better, by using resources from the internet/library ext, but improvement has been slow. It can be seen, but I still feel i am behind where i should be at my age, and for how long i have been drawing/painting.

I've met many artists who are self taught, and never went to college at all for what they do, yet i have it in my mind that college will help me greatly, since i learn faster when taught by another, rather then myself.

My question is, is this actually possible? do some people just have a harder time of teaching theirself? I have the dedication and interest to pursue education, but i respond so much better to being instructed, rather than instructing myself. Is this normal?

**Num beams = 1**. TL;DR: i want to continue my art education, but still be able to teach myself as much as possible, is it possible to do both at the same time? I appreciate all inputs.

**Num beams = 5**. TL;DR: 18, recently out of high school, and will be going to college for a bachelors in illustration in likely 1-2 years. Slow but steady improvement in my drawing/painting skills, but i respond better to being taught rather then instructing myself. Is this normal?,

**Num beams = 20**. TL;DR: 18 year old self taught myself to be better at drawing/painting over the past 2 years, but still feel i am behind where i should be at my age, and for how long i have been drawing/painting. Do some people just have a harder time of teaching themselves? Is this normal?

SUBREDDIT: r/legaladvice

TITLE: Mom was hurt at work, I think it's something that is a little past worker's comp. Any advice?

POST: My mom works at an old navy for something upwards of almost ten years. Over the span of her working their they have been lingering in and out of being abusive employers. Not that this isn't something I wouldn't expect out of a big corporation, something happened recently that I'm close to positive should be lawsuit worthy or something in my mom's favor.

The other day on the clock my mom was cleaning one of the bathrooms and while taking the garbage out there was a faced up syringe under some paper towels! She found out after jabbing her hand with the syringe and noticed that there was preexisting blood on this syringe!!! She was then brought to the hospital (obviously) to be screened for all sorts of diseases and had the option of starting intense medications to prevent the microscopic chance of that needle being contaminated with HIV though the medications would make her very ill. All old navy did about the situation was lock the bathrooms so the public can't use them.

Is there anything about this situation that my mom can use in her favor? She didn't turn the syringe over to Old Navy because they were being very nonchalant and dismissive about the situation.

**Num beams = 1**. TL;DR: mom is a working at old navy, had a bloody syringe found under some paper towels on the garbage disposal, likely contaminated with HIV/AIDS. Any help in what she can use in her favor in future lawsuits or otherwise?

**Num beams = 5**. TL;DR: Mom found syringe with preexisting blood under paper towels at work. Old navy was dismissive and locked the bathrooms so the public couldn't use them. Is there anything my mom can use in her favor? She didn't turn the syringe over to Old Navy because they were being very nonchalant and dismissive about the situation.

**Num beams = 20**. TL;DR: Mom found a syringe in the bathroom at work that had preexisting blood on it. Old navy was dismissive about the situation and locked the bathrooms so the public couldn't use them. Is there anything my mom can use in her favor? She didn't turn the syringe over to Old Navy because they were being very nonchalant and dismissive about the situation. A: My mom works at an old navy for something upwards of almost ten years. Over the span of her working their they have been lingering in and out of being abusive employers. Not that this isn't something I wouldn't expect out of a big corporation, something happened recently that I'm close to positive should be lawsuit worthy or something in my mom's favor. The other day on the clock my mom was cleaning one of the bathrooms and while taking the garbage out there was a faced up syringe under some paper towels! She found out after jabbing her hand with the syringe and noticed that there was preexisting blood on this syringe!!! She was then brought to the hospital (obviously) to be screened for all sorts of diseases and had the option of starting intense medications

SUBREDDIT: r/self

TITLE: Always seize the opportunity to help others

POST: This is my first post so here's a little intro about what I do for work. I'm a security guard. I work the graveyard shift in the downtown part of my city. Nothing special. Being a security guard is easy money.

Anyways, while making my first round I noticed a larger man in a wheelchair across the street trying to make it onto the curb. The transition for the handicapped access wasn't smooth enough and he was stuck. I crossed and pushed him onto the sidewalk.

He needed to go to the hospital five blocks the road. I called my supervisor and said I'd be back in a few I had to help this guy. I pushed him to the hospital and walked back.

If I had my headphones in like any other day, I wouldn't have seen him and he'd be struggling to get there.

I'm not sure what was wrong but he had one arm and one leg missing.

**Num beams = 1**. TL;DR: I helped a man who needed emergency attention on the street by pushing him for five blocks because I had headphones in. I didn't see him until after I had to walk back.

**Num beams = 5**. TL;DR: I saw a handicapped guy stuck on the sidewalk. I pushed him to the hospital. If I had my headphones in I wouldn't have seen him and he'd be struggling to get there. I'm not sure what was wrong but he had one arm and one leg missing.

**Num beams = 20**. TL;DR: I saw a handicapped guy stuck on the sidewalk. I pushed him to the hospital. If I had my headphones in, I wouldn't have seen him and he'd be struggling to get there. I'm not sure what was wrong but he had one arm and one leg missing.

### F.2   Samples from DPO with and without SFT Pre-Training

SUBREDDIT: r/Parenting

TITLE: Young adults/reddit parents: how have you found ways of asking a parent to loan you money and figure out a way of paying it back? Diplomatically.

POST: I know this question has a lot to do with the kind of relationship one has with each parent and at what stage in life you are personally and professionally.

I'm 27, moved two states away for grad school, and i'm living strictly off loans. It's my first semester and I'm taking 12 hours. I'm an ex-teacher, so I have some professional background. No savings. I've travelled quite a bit though. Shooting for an assistantship/internship in the Spring. I've never been this low on funds. It's also scary realizing how much deeper I am putting myself into the dark debt hole of student loans.

I'm pretty much living on a credit card for the next two months and I am too ashamed to ask my parents for any money. I can nanny here or there, but it takes time and effort (yes, like everything) to find families needing a babysitter in a giant college town. I will get organized enough to pay my father back, of course. He's already loaned me close to $1,000 in the past four or five months and I feel a little disgusting since he has a hard time making any income himself. Mother is another story–she's got a really tight grasp on her pocket and is a little aggressive when it comes to financial advice. So I never approach her.

My dad is the kindest man I know and has the cleanest reputation among his contacts, which is why people trust him and partner with him so much. His kindness does get the best of him however, sometimes to his detriment. In other words, people have walked all over him in the past. Mom, on the other hand, is constantly reminding me about payments, asking me about my loan, really approaching things in a way that push me far from wanting to face finances. Fabulous parenting. It's always been this way, with anything growing up. Dad never seemed to say no, he always gave me options/choices. Mom was a straightforward, dry, clear, British NO. She has always said I don't have my feet firmly grounded. I'm getting off on a tangent now so I'll stop.

**With SFT.** TL;DR: Since I inherited money and owe my father a significant chunk of it, my mother doesn't allow herself to see any sense in dealing with my finances. She constantly reminds me I need to pay my loan, yet seems to be a little aggressive and pushy about things. How do I confront that without making my mother uncomfortable?

**Without SFT.** TL;DR: [CHINESE SCRIPT]

SUBREDDIT: r/AskReddit

TITLE: [Serious] How can I tell my mother politely, that she is dealing with our dog in an inappropriate way?

POST: My mother watched Cesar Millan a lot before we got our dog, and she acted like she knew everything there was to know about training and raising a dog.

Its been about 2 years with our dog, and he has a problem with barking at the door when people go to open it (doesn't seem like that much of a problem to me). My mother's reaction is to yell at him, or if he does it sequentially, she will strike him in the thigh/rear end.

Whenever I step in and try and take control of the situation, she will get mad, and try and tell me how to handle this kind of thing, because she thinks she is a dog expert. (This upsets me because it feels like I am being talked down to)

The last strand was crossed today, when I was in the basement watching tv, I heard my dog bark at the door, and then my mother get up and yell at him. He ran into our sun room, which is above the basement, and I could hear her hit him, and his paws skid on the floor. I went up to check on him (he was fine, no damage, happy little guy when he's with me) and my mother yelled at me to put him back in the room where he is to stay ¨for a long long time¨because he tried to bite her when she was ¨disciplining¨him.

Her solution to my dogs problem is completely inappropriate, and no way to treat a dog. I believe in positive reinforcement, and I am getting very frustrated listening to her complain about my dog, and her inappropriate ways of dealing with his barking.

So if anyone knows a way I can tell her what she is doing is wrong, or if there is a way I can try and get my dog to stop barking at the door, please let me know.

**With SFT.** TL;DR: My mother's ways of dealing with our dog's barking problem is inappropriate and is getting worse. How can I tell her otherwise, or get her to change her behavior?

**Without SFT.** TL;DR: [CHINESE SCRIPT]

SUBREDDIT: r/legaladvice

TITLE: Speeding down highway. Can I get a ticket?

POST: 18, Male, Tennessee.

I was speeding, like 120 in a 65. Some girl kept trying to speed up with me in a shit car, I drive a turbo g35 it was no match but she wouldn't budge she'd push her little car around trying to pass me, and obviously putting her life in danger, I'm no godly driver but I know my car, and the road. I knew what I could do speed wise, and I constantly took it back to around 80 before she'd speed up. Eventually I had to turn, and she held up her phone, and a piece of paper, I assumed it was her number but now that I think about it she probably got my tags and plans to report or something, I want to know could I get in trouble?

On mobile.

**With SFT.** TL;DR: speeding 120 in a 65, girl tried to speed up with me in shitty car, had phone with street address on it and number for police, could get in trouble for following too close.

**Without SFT.** TL;DR: Female driver started tailgating, almost hitting me in my car. Then tried to make passes. I know my car works well, but she wanted to prove me wrong, even though I'm pretty sure I'd outrun her at a lower speed. She constantly kept slowing me down so I'd be at a speed she could pass me. But after getting

## G   Dataset and Hyperparameter Details

The TL;DR dataset contains 64,832 summary comparisons and is derived from the Webis TLDR dataset, with human feedback collected by OpenAI. Each example is scraped from Reddit and belongs to one of several "subreddits" (topic forums), with an associated title/post/human TL;DR (summary). Around 5% is held out for validation.

Each TL;DR model was a pre-trained Pythia 2.8B model. We first SFT with a learning rate of $0.5 \times 10^{-6}$ and batch size of 128 for one epoch using 8 gradient accumulation steps on 4 NVIDIA A40 GPUs. All DPO models with the same setup and hyperparameters as the original paper (Rafailov et al., 2023). Specifically, we train for 1 epoch with a learning rate of $0.5 \times 10^{-6}$ and linear warmup of 150 steps. We use a batch size of 32 (example example is a positive and a negative) and clip gradient norms to 10. We use $\beta = 0.1$ for DPO.

All generation samples are performed with temperature 1.0 and max length 512 unless otherwise specified. Evaluations (winrates, lengths, etc) are computed with 256 samples from the held-out set in TL;DR.

## H   GPT 4 Evaluation

We use the following prompt in all our GPT 4 evaluations. The order of the model sample and the reference response is randomized in each evaluation to avoid positional bias in the judge.

Which of the following summaries does a better job of summarizing the most  important points in the given forum post?

Post: QUERY

Summary A: A

Summary B: B

FIRST provide a one-sentence comparison of the two summaries, explaining which  you prefer and why. SECOND, on a new line, state only "A" or "B" to indicate your  choice. Your response should use the format:

Comparison: ¡one-sentence comparison and explanation¿ Preferred: ¡"A" or "B"¿

