# OpenReview forum: "From $r$ to $Q^*$: Your Language Model is Secretly a Q-Function"
_colmweb.org/COLM/2024/Conference — COLM_

### Official Review · Reviewer_Qos1 · 2024-05-11

**Rating:** 6
**Confidence:** 3
**Ethics Flag:** 1

**Summary:**

This paper formulates DPO in the token-level MDP and provides practical insights into DPO policy based on the derivation. The derivation is carefully step-by-step, and the paper is well organized. The paper provides interesting insights into the token-level behavior of DPO.

**Questions To Authors:**

* I do not clearly see the equivalence between Eq (2) and the formulation of RLHF in the previous studies. Are they exactly the same?
* The first line in Section 4.1 says, "While the original derivation of DPO relies on the fact that $Q^*(x, y) = r(x, y)$," but I could not find this statement before this line. Where is it explained?
* Isn't M and N opposite in Eq. (9)?
* Section 5.3 provides interesting findings, but what is its connection with the token-level formulation of DPO?
* Typos
  - Abstract: multi-tun -> multi-turn
  - Section 1, paragraph 4: searchgn -> searching; missing citation in the next line.
  - Section 3.1, after Eq (1): "loss" -> "lose"?
  - Section 3.3, paragraph 1: redundant "entirely"
  - Setion 4.1, paragraph 1: $\pi*$ -> $\pi^*$

**Reasons To Accept:**

* The token-level formulation provides a novel token-level interpretability of DPO.
* The derivation is well explained.
* The paper is well-organized and easy to follow.

**Reasons To Reject:**

* I understand that the main contribution is the token-level formulation of DPO, and I find the practical insights based on this formulation interesting. However, the empirical results are too weak to support the practical insights the paper insists on.
  - Section 5.1 shows only several qualitative results (including the appendix) to argue that DPO learns token-level rewards.
  - Section 5.2 attempts to make a connection between guided decoding and DPO by showing that DPO improves performance with moderate beam size. However, the relationship between the argument and the results is unclear. In text generation, it is well-known that beam search with a moderate size of beam improves performance on (e.g., [1]). Why should the cause of that improvement be attributed to DPO?
* The derivation provides a new interpretation of DPO but does not accompany improvement on DPO. This is a minor point and not the focus of this paper, but the current introduction makes such an expectation (e.g., "we rectify this difference").

[1] If beam search is the answer, what was the question? (EMNLP 2020)

---

> ### Author Rebuttal · Authors · 2024-05-28
>
> We would like to thank the reviewer for the detailed feedback! We already provided an answer about the evaluations in Section 5.1. in our response to reviewer UQbN and to your second question in our response to reviewer hRDL, which we will not repeat due to response length limitations.
>
> **“Section 5.2 attempts to make a connection between guided decoding and DPO by showing that DPO improves…”**
>
> A number of recent works ([1]-[3] for reference), have explored using search-based methods with a value function during inference time, which has shown to improve the quality of outputs. Moreover [1] has also shown such benefits when using a DPO base model as well. The goal of section 5.2 is to theoretically show the equivalence between such search methods and DPO likelihood-based search and provide empirical proof. As such we do not aim to explain the performance improvement of beam search, but to make the point that we do not need an explicitly trained separate value function.  Instead we can utilize the implicit value function learned by the DPO model, by leveraging the connection between implicit value functions and LLM likelihoods in DPO, which is the main theoretical contribution of our work as outlined in Section 4.
>
>
> **“I do not clearly see the equivalence between Eq (2) and the formulation of RLHF in the previous studies. Are they exactly the same?”**
>
> Prior classical RLHF works usually formulate the RLHF problem using bandit terms, however in practice the PPO algorithm (and similar policy-gradient based methods) optimize the token-level objective of Eq. 2.
>
> Thank you for pointing out a number of typos, we will fix these in our final version!
>
>
> [1] Controlled Decoding from Language Models, Sidharth Mudgal, Jong Lee, Harish Ganapathy, YaGuang Li, Tao Wang, Yanping Huang, Zhifeng Chen, Heng-Tze Cheng, Michael Collins, Trevor Strohman, Jilin Chen, Alex Beutel, Ahmad Beirami, 2024
>
> [2] Don't throw away your value model! Generating more preferable text with Value-Guided Monte-Carlo Tree Search decoding, Jiacheng Liu, Andrew Cohen, Ramakanth Pasunuru, Yejin Choi, Hannaneh Hajishirzi, Asli Celikyilmaz, 2024
>
> [3] Alphazero-like Tree-Search can Guide Large Language Model Decoding and Training, Xidong Feng, Ziyu Wan, Muning Wen, Stephen Marcus McAleer, Ying Wen, Weinan Zhang, Jun Wang, 2024

---

> > ### Comment · Reviewer_Qos1 · 2024-06-06
> >
> > I appreciate the authors' response.
> > I am already positive about the paper, but I have some concerns that the response does not sufficiently address.
> > * The concern about this Section 5.2 is not addressed in this response. In general, appropriately increasing the width of the beam search will result in better performance than the greedy search or excessively large beam size, and the fact that this phenomenon has been reconfirmed for DPO policy does not support that DPO is implicitly learning a value function. If one wants to make such a claim, it is necessary to conduct similar experiments and compare, for example, RLHF policy in the absence of value function guidance.
> > * The connection between Section 5.3 and the main claim, token-level formulation of DPO, remains unclear.

---

> > > ### Author Response · Authors · 2024-06-07
> > >
> > > **The concern about this Section 5.2 is not addressed…**
> > >
> > > We are working on evaluating a PPO-trained model under beam search for the same dataset and will report results as soon as they are available. As stated by the reviewer, a-priori we also expect performance gains in PPO policies with beam search. The goal of Section 5.2 is not to explain the performance gains of BS, but rather to show the equivalence of value-guided search in LLMs (which has driven a lot of research interest recently) to likelihood-based search (at least in the RLHF domain), of which BS is an example. We empirically show similar performance gains with simple BS as reported in prior works with value-guided decoding, without actually training a separate value function.
> > >
> > > **The connection between Section 5.3…**
> > >
> > > The expectation in Eq. 14 is implicitly taken over the sequence of tokens induced by the reference model, but we believe the reviewer is right and this result also holds in a bandit interpretation where the corresponding expectation is over the prompt distribution.

---

### Official Review · Reviewer_hRDL · 2024-05-12

**Rating:** 8
**Confidence:** 3
**Ethics Flag:** 1

**Summary:**

This work offers an alternative derivation of Direct Preference Optimization using Markov Decision Processes (MDPs) that enables to define token-level rewards. This has theoretical and practical applications. Fundamentally that DPO can learn token level rewards and assign different credit to tokens, a relation between guided decoding and search-based algorithms and an explanation of the phenomenon of decreasing likelihoods during DPO training.

The core of the paper is a derivation that follows closely that of DPO but from an MDP perspective i.e. [S4.1]

1. Determining the optimal policy for the KL-controled reward maximization loss. Compared to DPO here they consider a more general case of the loss that uses a MDP characterization with states and actions as a function of time. The optimal policy for this case is a know result, as it happened with DPO.

2. Deriving a the reward from the this optimal policy. Compared to DPO, here the use of the  Bellman Equations is needed to obtain the token level relations. The value function cancels in the same way the partition function cancels in DPO. I may have missed details on this part, but it seems solid. Result of this is Eq (9) a derivation of DPO as an MDP.

3. In addition to this they generalize the definition of equivalency of DPO for the MDP case. This exploits very similar properties as in the case of DPO (S4.2).

In the following sections they use these derivations to deduce certain properties of DPO and empirically confirm these i.e.

4. The MDP derivation of DPO, indicates that it can learn token level rewards.

5. Guided decoding that maximizes a reward, also based in the MDP formulation, is the same as maximum likelihood DPO optimization.

6. Explain the known phenomena that "the implicit rewards of both the chosen and rejected response decline, though the margin between them increases", which once is seen from the perspective of the MDP it is explainable.

**Questions To Authors:**

### S 4.1
>While the original derivation of DPO relies on the fact that Q∗(x, y) = r(x, y)

*Question 1:* Can you clarify this assertion? The original derivation in (Rafailov et al 2023) does not use MDPs or Q. It basically consists on two elements

-  derivation of the form of the optimal policy for the KL-controlled reward maximization objective (appendix A.1). This just needs algebraic manipulation of the original objective and checking the solution satisfies this and is unique.

- derivation of the DPO objective, which follows again by algebraic manipulation of the optimal distribution derived above and replacing this in the Bradly-Terry model.

I am aware that there are earlier derivations for the optimal policy, but this one seems simple and not need of the assumption above.

### S4.1
>This style of relationship was first explored by Garg et al. (2022) in imitation learning and later in Hejna & Sadigh (2024) for preference-based RL. However, these works require the use of a discount factor γ < 1 which is typically not used in RLHF.

*Question 2:* Could you expand on this difference? It seems to be and important aspect to judge the novelty. Maybe add some summary of the derivation used in those works in the Appendix A.

### S.1
>Second, we demonstrate that likelihood search over a DPO model is analogous to searchign over a reward function during decoding as done by contemporary works ().

*Question 3:* can you provide this missing reference?

### S 4.2

>Interestingly, in practice, the potential function Φ(st) represents the free parameter in the logits of the language model. An equal shift along all logits yields the same policy, but different Q-functions and corresponding rewards. The above Theorem proves that all of these are in the same equivalence class and induce the same set of preferences.

I fail to see why this is not self-evident. Scaling the logits by f(x) does not affect the probability distribution. Could you please clarify this?

**Reasons To Accept:**

This paper provided novel and relevant insights into DPO, in particular

1. a theoretical derivation token-level reward DPO bringing concepts from Markov Decision Processes.
2. showing that DPO maximum likelihood search is equivalent to decoding to maximize a reward function.
3. provide insight and empirical results showing the importance of the starting policy initialization in DPOs training.

**Reasons To Reject:**

Not really reasons to reject, but more clarity on some RL terms NLP people may not be fully familiar with. Thanks to RLHF's impact in LLMs, the average reader on this conference will be familiar with a lot of the material (I am one of those), but maybe some concepts may remain unfamiliar, such as, bandits, MDP and Q function.

I have to say I did not fully understand some of the claims and some aspects seem self-evident, I put these on the question section. Probably after discussion this should all be ok.

---

> ### Author Rebuttal · Authors · 2024-05-28
>
> We would like to thank the reviewer for the detailed review and kind words!
>
>
> **“While the original derivation of DPO relies on the fact that Q∗(x, y) = r(x, y)...”**
>
> Indeed, the original DPO work uses a bandit setting which can be represented as a trivial single-step MDP, which does not require any sequential considerations. In that setting the value function Q(x,y) is trivially r(x,y), but as we argue in later sections, this does not seem to represent the real sequential nature of language problems.
>
> **“Question 2: Could you expand on this difference? It seems to be and important aspect to judge the novelty”**
>
> Garg et al. (2022) is an imitation learning algorithm, while we focus on the problem of improving language models from (offline) user feedback. Hejna & Sadigh (2024) trains a parameterized value function using the feedback dataset and then separately attempts to extract a policy from that value function. All experiments are based on continuous control and the approach has been hard to scale to LLMs. In our work we have a unifying formulation through max-entropy RL, where we represent the LLM itself as both a policy and an implicit value function and do not use separately parameterized models. Moreover, the proofs in those works state something similar, but critically different: that there is a bijection between optimal rewards and value functions under a fixed policy. Both our setting and results are different. We show that there is a bijection between the reward function and *optimal* Q-function (not Q-function of a fixed policy) and that this holds in the Token-MDP with discount factor one (while prior works require discount factors less than one in order to invoke contraction properties of the Bellman operator). This requires a different proof technique than used in prior work.
>
>
> **“I fail to see why this is not self-evident. Scaling the logits by f(x) does not affect the probability distribution. Could you please clarify this?”**
>
> This comment links the theoretical reward-smoothing formulation of Ng et. al. (1999) as represented in Section 4.2 to the practical LLM implementation. We agree that from the logit's perspective, it is clear that a constant shift does not affect the final model outputs, but represents the reward shaping potential $\Phi(s)$. However, from a token-level RL perspective this was originally non-obvious until Ng's. work. Here we extend this reward shaping formulation to the feedback setting, which we state in Theorem 1.

---

> > ### Comment · Reviewer_hRDL · 2024-05-31
> >
> > I thank the authors for their answers. Unless I missed something, this question remains unanswered
> >
> > >Question 3: can you provide this missing reference?

---

> > ### Author Response · Authors · 2024-05-31
> >
> > Hi! Sorry about that oversight. The references are:
> >
> > Liu, Jiacheng, et al. "Making ppo even better: Value-guided monte-carlo tree search decoding." arXiv preprint arXiv:2309.15028 (2023).
> >
> > Feng, Xidong, et al. "Alphazero-like tree-search can guide large language model decoding and training." arXiv preprint arXiv:2309.17179 (2023).
> >
> > Please let us know if you have any other questions!

---

> > > ### Comment · Reviewer_hRDL · 2024-06-05
> > >
> > > thanks for the refrences

---

### Official Review · Reviewer_UQbN · 2024-05-13

**Rating:** 7
**Confidence:** 4
**Ethics Flag:** 1

**Summary:**

This paper compares the classic pipeline adopted in the GPT series with DPO, a more recent method proposed by the research to overcome the complexities and intricacies of the RLHF pipeline with PPO. The main issue with DPO is that it is formulated as a contextual bandit problem where the entire response is considered as an arm ignoring the fine-grained decision-making process underlying language modelling and its relationship with credit-assignment.

As a first contribution, the authors demonstrate that it is possible to reframe DPO in the token-level MDP which gives the ability to shed some light on the credit assignment ability of this method. Additionally, because it is now possible to define a Q-function in terms of single tokens, it is possible to formulate search-based algorithms (such as Monte-Carlo Tree Search) via likelihood-based search implemented as beam-search. Finally, thanks to this theoretical framework, the authors study a well-known problem in DPO training which causes implicit rewards to decline during training and conclude that this is a result of the DPO training process (i.e., this is a feature of DPO and not a bug).

**Questions To Authors:**

## Suggestions regarding weaknesses of the case studies

1. [Section 5.1 - training details] The authors specify that they used Pythia as a base model but it's not clear how they completed the training for the Reddit TL;DR dataset. Was this model instruction-tuned first using SFT or not? Did they try completing this step for other open-source models? I believe it would be good to comment on this to provide more information regarding how their work will influence the overall LLM training pipeline.
2. [Section 5.1 - credit assignment comparison] I believe that the authors should execute this evaluation in a more systematic way. They only report very specific examples for the DPO model only. Does a model trained only with SFT behave in a similar fashion? Is there any difference between the two?
3. I would recommend the authors use the appendix to report more details regarding the training steps required to reproduce their case studies. I think it's important to at least include hyperparameters and other steps required to reproduce the training pipeline that they have used.

## Minor edits

1. Typo at Page 4: belmman equaitons
2. I would suggest the authors consider moving their related work at the end of the paper considering that the "Preliminaries" section provides a very good overview of all the technical background required to understand their theoretical contribution.
3. Authors should consider citing older work on using RL for image captioning such as [1]

## References
[1] Marcella Cornia, Matteo Stefanini, Lorenzo Baraldi, and Rita Cucchiara. 2020. Meshed-memory transformer for image captioning. In Proceedings of the IEEE/CVF Conference on Computer Vision and Pattern Recognition, pages 10578–10587.

**Reasons To Accept:**

1. Really well-written paper with a strong theoretical foundation that clarifies many of the issues around DPO and RLHF training
2. This paper represents a must-read for many practitioners and researchers interested in training LLMs but I believe it can extend further to other applications as well. In fact, the authors highlight LLM agents as a promising avenue for this work because in this domain the agent has to generate very long trajectories and sources for credit assignment are limited considering that they might happen only at the end of the trajectory (i.e., when the task is finished).

**Reasons To Reject:**

Overall, I think this paper represents a great contribution to the community and in general can be considered as a great work for many to learn from. However, I was a bit disappointed by the level of detail of the case studies reported in Section 5 which I will list below. Additionally, I will report more details and suggestions for authors in the "Questions for Authors" box.

1. [Section 5.1 - training details] Section 5.1 reports a case study on text summarization to study the ability of DPO to learn credit assignment but it somehow lacks some details regarding the concrete steps involved in training this model.
2. [Section 5.1 - credit assignment comparison] Again connected with the previous point, the evaluation feels very anecdotical. I believe this credit-assignment evaluation is valuable but it should be executed in a more principled way.
3. Overall, all the sections involving some training should report more details to facilitate other researchers to reproduce your experiments. I understand that the space is limited and authors decided to prioritise their theoretical contribution but I think it's important to report these details in the appendix.

---

> ### Author Rebuttal · Authors · 2024-05-28
>
> We would like to thank the reviewer for the detailed feedback and kind words!
>
> **“Section 5.1 reports a case study..”**
>
> We follow the same training procedure as original DPO work, including SFT pre-training and DPO fine-tuning including the official codebase and default hyperparameters, besides varying the beta coefficient, which we report in our results.
>
> **“Again connected with the previous point, the evaluation feels very anecdotical…”**
>
> We appreciate your point! Note that we have included additional examples in the Appendix. Furthermore, additional evaluations on TL;DR have exhibited similar trends -- we can include more of these examples in the final paper.
>
> We believe larger scale studies are indeed warranted, but face challenges:
>
> 1. Quantifying credit assignment in reinforcement learning in general is challenging, but becomes increasingly so in free-form text generation, where overall quality might not be attributable to specific tokens but larger scale features of the writing.
> 2. Datasets for such analysis are quite limited.
>
> One potential domain where credit assignment might be easier to evaluate is multi-step reasoning tasks. Recent work [1], trains a DPO implicit reward model as a verifier for sequence-level reasoning. It is possible to compare step-level results from such a verifier to data with explicit step-level annotations [2]. We believe this is a valuable study, but in of itself can be quite expansive work to correctly train such models (which has warranted several publications in its own right) and is beyond the scope of our current work!
>
> We would appreciate it if the reviewer has any particular suggestions on this front!
>
> **“Did they try completing this step for other open-source models...”**
> We only considered the 2.8B Pythia model due to compute constraints, but we believe our results will transfer to stronger models as well.
>
> **“Does a model trained only with SFT behave in a similar fashion…”**
>
> Our credit assignment experiments all evaluate the DPO implicit reward, as outlined in Section 4.2. As such in of itself the SFT model does not induce a directly comparable reward function.
>
>
> [1] V-STaR: Training Verifiers for Self-Taught Reasoners, Arian Hosseini, Xingdi Yuan, Nikolay Malkin, Aaron Courville, Alessandro Sordoni, Rishabh Agarwal, 2024
>
> [2] Let's Verify Step by Step, Hunter Lightman, Vineet Kosaraju, Yura Burda, Harri Edwards, Bowen Baker, Teddy Lee, Jan Leike, John Schulman, Ilya Sutskever, Karl Cobbe, 2024

---

> > ### Comment · Reviewer_UQbN · 2024-06-04
> > **Response**
> >
> > I thank the authors for your honest answers. I would appreciate it if they could report the points that I raised above as limitations of this work which, nevertheless, I still believe represents a very good contribution to this conference.

---

> > > ### Author Response · Authors · 2024-06-05
> > >
> > > Thank you! We will be sure to report the points raised as limitations.
> > >
> > > 1. We will report full details on implementation. All of our hyper-parameters were taken from the official DPO repository. For sake of completeness, we include this here. We use Pythia 2.8B for all experiments. We first SFT on the preferred responses using batch size 64 with two gradient accumulation steps and learning rate of 5e-7. Prompt-responses are truncated to 512 tokens and we train for one epoch using the RMSProp optimizer. Gradient norms are clipped to 10. We then run DPO using batch size 32 (each example is positive and negative) again with 2 gradient accumulation steps. We use the same learning rate and truncation for DPO. For DPO we use beta = 0.1.
> > >
> > > 2. We will qualify the credit assignment portions of the draft to further indicate that we are using qualitative (not quantitative) analysis and that the existing evidence is anecdotal in Section 5.1.
> > >
> > > We hope this addresses your suggestions!

---

### Decision · Program_Chairs · 2024-07-10

**Decision:**

Accept

**Comment:**

This theoretical paper sheds new lights into connections with the status quo algorithm of RLHF (PPO) and a popular alternative introduced not so long ago (DPO). In particular it provides an alternative derivation of Direct Preference Optimization that shows that it does produce token-level rewards (instead of single final action at the end of the sequence).

This alternative derivation then allows the authors to leverage beam search for decoding and also to explain the unsatisfactory phenomena where the rewards of both completions (better & worse) decline all while the difference between them increases.

Most reviewers loved this paper, despite some experimental conclusions which are not of the same level than the theoretical insights. It would certainly be a valuable addition to COLM's programme and trigger insightful discussions.